# Analysis of the Nutritional Composition of Ready-to-Use Meat Alternatives in Belgium

**DOI:** 10.3390/nu16111648

**Published:** 2024-05-27

**Authors:** Evelien Mertens, Peter Deriemaeker, Katrien Van Beneden

**Affiliations:** 1Department of Health Care, Design and Technology, Nutrition and Dietetics Program, Erasmushogeschool Brussel, 1090 Brussels, Belgium; peter.deriemaeker@ehb.be (P.D.); katrien.van.beneden@ehb.be (K.V.B.); 2Department of Movement and Sport Sciences, Faculty of Physical Education and Physiotherapy, Vrije Universiteit Brussel, 1050 Brussels, Belgium

**Keywords:** plant-based, vegetarian, vegan, meat alternative, protein source

## Abstract

Background: The interest in meat alternatives has increased over the years as people embrace more varied food choices because of different reasons. This study aims to analyse the nutritional composition of ready-to-use meat alternatives and compare them with meat (products). Methods: Nutritional composition values were collected in 2022 of all ready-to-use meat alternatives in Belgian supermarkets, as well as their animal-based counterparts. A one-sample *t*-test was performed to test the nutritional composition of ready-to-use meat alternatives against norm values, while an independent samples *t*-test was used to make the comparison with meat. Results: Minced meat and pieces/strips/cubes scored favourably on all norm values. Cheeseburgers/schnitzels, nut/seed burgers and sausages contained more than 10 g/100 g total fat. The saturated fat and salt content was lower than the norm value in each category. Legume burgers/falafel contained less than 10 g/100 g protein. Vegetarian/vegan minced meat and bacon contained fewer calories, total and saturated fat, and more fibre compared to their animal-based counterparts. Conclusions: Minced meat and pieces/strips/cubes came out as the most favourable categories regarding nutritional composition norm values. Vegetarian/vegan steak came out the least favourable compared to steak, while vegetarian/vegan minced meat and vegetarian/vegan bacon came out the most favourable compared to their animal-based counterparts.

## 1. Introduction

The interest in meat alternatives has increased over the years as people embrace more varied food choices because of ecological, economical, religious, ethical and health reasons [1,2,3,4,5]. Food choices can be categorised into food patterns according to the number of animal-based products included. A vegetarian food pattern is absent of any type of meat, including poultry, seafood, fish and products containing them. In an ovo-lacto-vegetarian food pattern, milk and dairy (products), as well as eggs, are included, while in a lacto-vegetarian food pattern, only milk and dairy (products) and, in an ovo-vegetarian food pattern, only eggs are included. In a vegan food pattern, any type of food of animal origin is excluded [6]. In Belgium, one in three adults indicates that they eat vegetarian at least once a week. In parallel, the number of Belgians who eat meat and fish every day continues to decline [7]. An important incentive for the transition towards more plant-based protein sources is the impact of animal-based protein sources on the planetary boundaries. The current food production system is responsible for more than 25% of the global greenhouse gas emissions and causes deforestation, as large amounts of water, land and other natural resources are required for the food production of animal-based protein sources [5,8,9]. According to the EAT-Lancet report, a food pattern rich in plant-based foods and less animal-based protein sources has a positive impact on planetary and human health [10]. Thus, a higher proportion of healthy plant-based foods is protective against several chronic non-communicable diseases such as diabetes type 2, cardiovascular diseases, several types of cancer and chronic kidney disease [11,12,13,14,15,16,17,18]. National and international (policy) initiatives, such as the Green Deal, encourage the consumption of plant-based foods [10]. Yet changing dietary behaviour remains a challenge as taste preferences, culinary traditions, food neophobia as well as social and cultural norms often outweigh the perceived benefits of plant-based foods [19,20]. Studies indicate that legumes and meat analogues are most accepted as alternative protein sources, while insects and cultured meat are least accepted [21]. The 2014 Belgian Food Consumption Survey shows that the Flemish population consumes, on average, only 4 g of vegetarian products (both unprocessed or minimally processed vegetarian products (such as legumes, tofu, tempeh, and seitan)) and more processed ready-to-use meat alternatives (such as vegetarian/vegan burgers, vegetarian/vegan minced meat) per day [22]. The ready-to-use meat alternatives resemble meat in texture, mouthfeel, taste and appearance, appealing to an increasing number of consumers [23]. Consequently, their popularity has increased in recent years: in Belgium, sales increased by 24% between March 2020 and February 2021 [24]. In terms of food product development, plant-based protein sources are on the rise as worldwide meat alternatives are more often included in both a (partly) plant-based and an omnivore food pattern [25]. Many ready-to-use meat alternatives are considered ultra-processed using the NOVA classification (a classification that classifies products by degree of processing), which may delay their acceptance by consumers because, in most studies, ultra-processed foods are associated with adverse health effects, such as overweight and obesity, cardiovascular disease and overall mortality [26]. In general, ultra-processed foods are defined as foods produced with a small content of whole foods or with no whole foods at all but with processed ingredients or substances that come from whole foods (for example protein isolates, oils, hydrogenated fats, flours and starches, several types of sugars and refined carbohydrates, etc.) [27,28]. However, the question rises whether various types of ultra-processed foods contribute differently to the risk of developing non-communicable diseases and multimorbidity. According to a multinational cohort study plant-based alternatives were not associated with a risk of multimorbidity, in contrast to other types of ultra-processed foods such as for example animal-based products and artificially and sugar-sweetened beverages [29]. Nuance is incredibly important to distinguish between ultra-processed foods that carry negative health effects and (ultra-) processed foods with a higher nutrient density [30,31,32]. The threshold to cook with ready-to-use meat alternatives is small for many consumers because (1) ready-to-use meat alternatives can have better digestibility than, for example, beans, (2) more variety is possible, (3) they have higher protein content and higher bioavailability of protein than legumes, and (4) they have the same functional properties (in terms of use, appearance and method of preparation) as meat. This facilitates the transition to more plant-based protein sources [33,34]. Individuals opting for a vegetarian or vegan food pattern do not need meat alternatives that resemble the organoleptic and sensory properties of meat as much, while omnivores and flexitarians prefer meat alternatives that approximate meat in all respects [35,36,37]. According to the literature, the primary drivers of food choice are taste, healthfulness, price/cost and convenience [38,39]. Studies indicate that current ready-to-use meat alternatives can offer similar nutritional values to meat [40]. Ready-to-use meat alternatives, compared to animal-based protein sources, tend to have lower levels of total fat, saturated fat, cholesterol, and total calories, but may contain less (bioavailable) protein, iron, and vitamin B12. Although plant-predominantly and totally plant-based food patterns are appropriate for all stages of the life cycle (including pregnancy and lactation, childhood, at older ages, and for athletes), these points of interest should be taken into account [1,6]. A well-balanced plant-based food pattern and the regular use of fortified foods and/or supplements should be pursued [1,6]. Although some ready-to-use meat alternatives contain large amounts of salt, they also tend to be higher in fibre and several micronutrients [41,42,43,44,45,46,47,48,49].

The range of products available in supermarkets is increasing, so consumers often do not know which products are the most beneficial according to health. For the time being, no studies are available on the nutritional composition, Nutriscore, Ecoscore and price of uncooked/unprepared ready-to-use meat alternatives and the comparison of the equivalent meat (products) on the Belgian market. The present study aims to analyse the nutritional composition and compare them with the norm values of macronutrients and micronutrients (developed by the Belgian Professional Association of Dietitians) of vegetarian and vegan ready-to-use meat alternatives applicable in Belgium. A second aim of the present study is to compare the nutritional composition of ready-to-use meat alternatives to their animal-based counterparts. It remains extremely important to consider product by product, but there is a greater chance of making a nutritionally higher-quality choice taking into account the information from the present study.

## 2. Materials and Methods

### 2.1. Ready-to-Use Meat Alternatives

A database of the nutritional composition (calories, protein, total fat, saturated fat, fibre, salt, iron and vitamin B12), ingredients, price, etc., of all uncooked/unprepared ready-to-use meat alternatives (i.e., no legumes, as well as no tofu, tempeh, seitan) sold in Belgian supermarkets was established between May and December 2022 in Microsoft Excel. The inclusion criteria were vegetarian (partially plant-based with one or more animal-based components such as egg and/or milk/cheese) and vegan (totally plant-based) fresh or frozen ready-to-use meat alternatives sold in Belgian supermarkets. For the comparison with the norm values, the ready-to-use meat alternatives were split into categories based on their name and first/main ingredient(s). Nuggets were grouped under schnitzels/breaded burgers, and bacon strips were grouped under pieces/strips/cubes, leading to 13 categories of ready-to-use meat alternatives, which were compared to the norm values. Regarding the comparison with meat products, 10 categories of products were made based on their name and similar characteristics with comparable meat (products).

Alongside nutritional composition and price, the Nutriscore and Ecoscore were also gathered.

The Nutriscore is also a visual representation of nutritional value, which consists of a colour and a letter. That combination indicates which foods within a particular product group are healthier than others. The colour and letter are determined by the algorithm developed by the French Nutritional Epidemiology Research Team. The algorithm pairs positive attributes (content of protein, fibre and fruits, vegetables, nuts) with negative attributes (content of energy, sugars, saturated fat, sodium) to arrive at a score between −15 (best choice) and +40 (least good choice). That score is reduced to a combination of a letter (A to E) and a colour (from dark green to red). Dark green represents the best nutritional value (preferable), and red represents the worst nutritional value (to be avoided).

The Ecoscore is a visual representation of ecological value that also consists of a colour and a letter. Firstly, the environmental impact of a product throughout its life cycle is analysed. This results in a score out of 100, to which plus and/or minus points are assigned via the bonus-malus system based on additional indicators such as origin and packaging. This final score is translated into an Ecoscore from A to E, where A is the most ecological product, and E is the least ecological product.

### 2.2. Meat (Products)

Between May and December 2022, a database of the nutritional composition (calories, protein, total fat, saturated fat, fibre and salt) was compiled in Microsoft Excel with meat (products) in Belgium supermarkets in order to make the comparison between ready-to-use meat alternatives and meat products. Meat products were categorised into 10 groups based on the type of meat product.

### 2.3. Statistical Analyses

Statistical analyses were performed using SPSS 28.0 (SPSS Inc., Chicago, IL, USA) with a significance level of 0.05. Descriptive statistics such as mean, standard deviation, minimum and maximum of the nutritional composition, as well as the price (only for the ready-to-use meat alternatives) were calculated per 100 g of product. The normality of the data was checked using the Kolmogorov–Smirnov test. Parameters that were not normally distributed were analysed non-parametrically. A non-parametric one-sample *t*-test was performed to test the nutritional composition against the norm values of protein (≥10 g/100 g), total fat (≤10 g/100 g), saturated fat (≤5 g/100 g), salt (≤1.625 g/100 g), iron (>0.7 mg/100 g) and vitamin B12 (>0.13 µg/100 g), which were developed by the Belgian Professional Association of Dietitians [50]. In terms of norm values, Nutriscore and Ecoscore were compared with a score of ‘A’ (number 1 in the one-sample *t*-test). A one-way ANOVA with Tukey post hoc test was performed to check the difference in price between the different types of ready-to-use meat alternatives. A non-parametric independent samples *t*-test was performed to compare ready-to-use meat alternatives with meat products.

## 3. Results

Table 1 shows the general characteristics/information of the ready-to-use meat alternatives. This table shows the categories of ready-to-use meat alternatives and the breakdown by percentage of vegan, private label, gluten-free, frozen and organic ready-to-use meat alternatives; 88.9% of minced meat is vegan, while only 9.1% of cheeseburgers/schnitzels are vegan. Half of the nut/seed burgers are from a private label, while for sausages, this is only 18.8%. Minced meat (40.7%) and pieces/strips/cubes (40.3%) have a high proportion of gluten-free variants, while there are no gluten-free variants within the breaded vegetable burgers/balls and cheeseburgers/schnitzels. 18.9% of the breaded vegetable burgers/balls are frozen, unlike the cheeseburgers/schnitzels, (pseudo)grain burgers and nut/seed burgers, where no frozen variant is available. Next, 86.7% of (pseudo)grain burgers are organic, while a very small proportion of schnitzels/breaded burgers/nuggets (6.6%) and hamburgers/chicken burgers (6.7%) are organic.

Table 2 shows the differences in nutritional composition compared to the norm values. In terms of protein content, minced meat, schnitzels/breaded burgers/nuggets, hamburgers/chicken burgers, steak, pieces/strips/cubes, sausages, and meatballs have a statistically significant higher protein content, while legume burgers/falafel have a statistically significant lower protein content than the norm of 10 g of protein per 100 g. All types of minced meat and steak included in the database achieved the norm, while only 15% of legume burgers/falafel and 20% of (pseudo)grain burgers approached the norm of 10 g of protein per 100 g.

Minced meat, vegetable burgers/balls (not breaded) and pieces/strips/cubes have a total fat content below the norm value of 10 g per 100 g. Cheeseburgers/schnitzels, nut/seed burgers and sausages have a statistically significant higher fat content than the norm of 10 g of total fat per 100 g.

All 13 categories have a statistically significant lower saturated fat content than the norm value of 5 g per 100 g, as well as a statistically significant lower salt content than the norm value of 1.625 g per 100 g. Only a small percentage of ready-to-use meat alternatives have higher saturated fat and salt content than the norm. The average salt content of pieces/strips/cubes is overestimated because the alternatives to bacon strips (which contain more salt than other types of pieces/strips/cubes) were included in this category.

Iron and vitamin B12 were only added to the minority of ready-to-use meat alternatives, but when it was added, the mean values exceeded the norm of 0.7 mg/100 g iron and 0.13 µg/100 g vitamin B12 in all 13 categories. Within each category, all fortified products achieved the norm.

Table 3 shows the differences in Nutriscore and Ecoscore compared to the norm value and the differences in price per 100 g of product. The Nutriscore and especially the Ecoscore are only known in a minority of ready-to-use meat alternatives. Seven categories (breaded vegetable burgers/balls, cheeseburgers/schnitzels, (pseudo)grain burgers, hamburgers/chicken burgers, pieces/strips/cubes, sausages and meatballs) have a statistically significantly less favourable score than Nutriscore A. In terms of Ecoscore, sausages have a statistically significant higher score than Ecoscore A.

In terms of price per 100 g of product, nut/seed burgers are the most expensive category, whereas minced meat, schnitzels/breaded burgers, unbreaded vegetable burgers/balls, breaded vegetable burgers/balls, legume burgers/falafel, hamburgers/chicken burgers and meatballs are slightly cheaper in terms of price per 100 g of the product.

Table 4 shows the main ingredients (top five protein and oils/fat sources) of the different categories of ready-to-use meat alternatives. Soy protein was the most selected protein source in vegetarian and vegan minced meat, schnitzels/breaded burgers/nuggets, vegetable burgers/balls (not breaded), hamburgers/chicken burgers, steak, chunks/strips/cubes, sausages and meatballs. In general, sunflower and rapeseed oil were the most commonly used oils/fat sources.

Table 5 shows the differences in nutritional composition (calories, protein, total fat, saturated fat and salt) between ready-to-use meat alternatives and meat products. Vegetarian/vegan minced meat scores statistically significantly lower in calories, total fat and saturated fat and higher in fibre compared to the animal-based counterpart. Vegetarian/vegan schnitzels/bread burgers/nuggets contain statistically significantly less protein and saturated fat and more fibre than the animal-based counterpart. The vegetarian/vegan cheeseburgers/schnitzels have a statistically significant lower protein content and a higher fibre content compared to cheeseburgers/schnitzels with a meat component. Vegetarian/vegan hamburgers have a statistically lower protein content and a higher fibre content compared to hamburgers. Vegetarian/vegan chicken (pieces) (unbreaded and without marinade) contain statistically significantly less protein and saturated fat and more fibre and salt compared to chicken (pieces) (unbreaded and without marinade). Vegetarian/vegan chicken (pieces) (breaded/with marinade) contain statistically significantly less protein and saturated fat and more fibre and salt than chicken (pieces) (breaded/with marinade). Vegetarian/vegan steak contains statistically significantly more calories, total fat, fibre and salt and less protein compared to steak. Vegetarian/vegan gyros/shoarma/pita meat has statistically significantly less saturated fat and more fibre and salt compared to gyros/shoarma/pita meat. Vegetarian/vegan bacon has a significantly lower value of calories, total fat, saturated fat and salt and a higher value of fibre compared to bacon. Vegetarian/vegan sausages have statistically significantly less protein and saturated fat and more fibre and salt compared to sausages.

A Appendix A in Microsoft Excel was added with the nutritional composition, Nutriscore, Ecoscore, price and brand of the ready-to-use meat alternatives and the nutritional composition of meat (products).

## 4. Discussion

The first aim of the study was to analyse the nutritional composition, Nutriscore, Ecoscore and price of ready-to-use meat alternatives available in Belgian supermarkets and, if possible, compare them with the norm values which were developed by the Belgian professional association of dietitians. The results show that minced meat, as well as pieces/strips/cubes, scored favourably on all norm values of macronutrients and micronutrients. Cheeseburgers/schnitzels, nut/seed burgers and sausages contain more total fat than the norm value of 10 g per 100 g product. The saturated fat content is lower than the norm value in each category, as is the salt content. Legume burgers/falafel scored less favourably than the norm value for protein and showed no statistically significant difference with the norm value of total fat. In terms of Nutriscore, legume burgers/falafel seem to be a healthy ready-to-use meat alternative according to their Nutriscore, but the protein content is too low, and other variants scored more favourably than the norm values in terms of total fat content. These results show that for ready-to-use meat alternatives, the Nutriscore should be interpreted with some caution, as it does not always guide the consumer to the most favourable ready-to-use meat alternative when taking into account the norm values developed by the Belgian Professional Association of Dietitians. Hopefully, with the adaptations in the calculation of the Nutriscore, there would be more synergy between Nutriscore and norm values of protein, total fat, saturated fat, salt and the fortification with iron and vitamin B12. When iron or vitamin B12 was added to a ready-to-use meat alternative, the levels exceeded the norm values. One explanation for the fact that most of the ready-to-use meat alternatives meet the norm value for protein is that a concentrated protein source (e.g., soy protein, wheat protein, pea protein, etc.) is used mostly in product development. Legume burgers/falafel scored significantly lower than the norm value for protein because, in many cases, they have unprocessed chickpeas (i.e., as a non-concentrated protein) as the first protein source. Although chickpeas have many health benefits, their protein content is limited. Cheeseburgers/schnitzels and nut/seed burgers contained more total fat than the norm value, with nut/seed burgers containing unsaturated fat especially. Although the type of fat in cheese is saturated, research shows that it is not really atherogenic [51]. The high total fat content in cheeseburgers/schnitzels can be explained by the fact that both milk or cheese and oils/fats are added to the product. All ready-to-use meat alternatives scored significantly lower than the norm value for saturated fat. This can be explained by the fact that the majority of products contain unsaturated oils/fat sources, mainly sunflower oil and rapeseed oil, while coconut oil, palm fat and shea butter are only found in a limited number of products.

A second aim was to compare the nutritional composition of ready-to-use meat alternatives to meat (products). A lot of vegetarian/vegan ready-to-use meat alternatives showed a (slightly) lower protein content compared to the meat (products), but the fibre content was higher and often, the saturated fat content was lower. However, in the Belgian population, protein intake is mostly adequate, while fibre intake is below the recommendation of 25 to 30 g per day [22,52]. Salt content is higher in five vegetarian/vegan categories compared to meat (products), but in ready-to-use meat alternatives, salt is already added by the producer, while the consumer mostly adds some extra salt when preparing meat (products).

There is still uncertainty around the health effects of ready-to-use meat alternatives compared to meat (products), as unprocessed and minimally processed plant-based protein sources such as legumes, tofu, tempeh and seitan are recommended because of their health benefits [53]. Epidemiological studies have shown health benefits from regular consumption of legumes. In meta-analyses of prospective observational studies, the consumption of legumes is associated with anti-carcinogenic properties (mainly for colon, prostate, stomach and pancreatic cancer), cardiovascular protection (both blood pressure and blood lipid values), reduction/delay of the ageing process, improvement of immune response, protection against type 2 diabetes, weight management, protection against osteoporosis, protection against gastrointestinal diseases and psychological health benefits [54,55,56,57,58,59,60,61,62,63]. The mechanisms behind these health benefits are multiple: (1) legumes contain complex carbohydrates with a low glycemic index, (2) they contain antioxidant properties [64], (3) the fibre serves as a food source and during the fermentation process of fibre by the intestinal bacteria secrete short-chain fatty acids [65], (4) they contain pectin, a fibre that reduces LDL cholesterol [65], (5) typical nutraceutical properties of legumes are attributed to bioactive substances such as (a) polyphenols (antioxidant, antimicrobial and anti-inflammatory properties [66,67]), (b) alkaloids (anti-carcinogenic activities and possess the ability to improve blood circulation in the brain [66]), (c) phytates (antioxidant [68]), saponins (beneficial influence on cholesterol levels and antimicrobial, antioxidant and anti-carcinogenic properties [66,69,70]), (d) enzymatic amylase inhibitors (inhibition of the digestive enzyme alpha-amylase, which prevents complex carbohydrates from being converted into simple carbohydrates such as glucose [71]), (e) lectins (anti-carcinogenic properties, stimulating the immune system, binding to tumor cell membranes, reducing cell proliferation and induce apoptosis, antimicrobial and insecticidal mechanisms [72,73]), (f) storage proteins and small peptides (exert hormone-like activities) [74]. Protein-rich soy products such as tofu and tempeh can reduce total serum cholesterol and LDL cholesterol but could also be effective in attenuating the effects of type 2 diabetes, blood pressure and cancer-related issues [75,76]. The composition and nutritional profile of seitan, which is low in (saturated) fat and calories, contains complex carbohydrates and is high in plant protein, helps in bowel movement and can lead to an increase in gut microbiota diversity, a decrease in serum cholesterol levels, a reduction in postprandial blood glucose level and a decreased risk of cancer and colitis [77].

Many ready-to-use meat alternatives are ultra-processed according to the NOVA criteria [28]. However, studies argue that the mere industrial processing of ingredients of plant origin does not make a product ultra-processed by default [30,31,32,78]. Mostly, the processing of plant-based ingredients may improve their nutritional profile [79]. The processing of legumes into ready-to-use meat alternatives can denature naturally occurring antinutrients and improve (protein) digestibility [79]. It is important to unravel the mechanism(s) by which ultra-processed foods may influence the risk of chronic diseases such as cardiovascular disease and diabetes type 2. Ultra-processed foods often have a higher caloric density combined with an altered food matrix, which leads to a softer texture that requires less chewing and delays satiety signalling [29]. The results of a study investigating the differences in satiety between a plant-based mince pasta meal and an equivalent meal with beef mince indicated that the pasta meal containing plant-based mince was more satiating than an equivalent meal prepared with beef mince. This was not associated with greater energy intake at a subsequent meal occasion [80]. However, previous studies showed that ready-to-use meat alternatives are often high in saturated fat and salt [41,81,82]. In the present study, 93% and 87.9% are in agreement with the norm values of saturated fat and salt, respectively. Presumably, the nutritional composition of these products has been adjusted favourably over the years. Although little research is available regarding the health effects of ready-to-use meat alternatives, while unprocessed and minimally processed plant-based protein sources such as legumes and derivatives are recommended, ready-to-use meat alternatives do offer some advantages over (mostly red and processed) meat. Ready-to-use meat alternatives, unlike meat, usually contain fibre, which has a beneficial effect on risk reduction of several non-communicable diseases, such as cancer, cardiovascular diseases, gastrointestinal disorders and type 2 diabetes [22,83]. Furthermore, red and processed meat consumption is associated with an increased risk of type 2 diabetes, cardiovascular disease and several types of cancers [84,85,86]. According to the literature, the increased health risk of consuming (especially red and processed) meat is due to one component or a combination of several mechanisms which is/are not found in plant-based protein sources. These components/mechanisms involve (1) saturated fat, (2) sodium content, (3) nitrate, (4) heme iron, (5) *N*-nitroso compounds, (6) heterocyclic amines and polycyclic aromatic hydrocarbons and (7) Neu5G [87,88,89]. Except for chicken (pieces) unbreaded and without marinade, 8 out of the 10 categories of meat (products) in the present study are ultra-processed, which is associated with health disadvantages such as an increased risk of type 2 diabetes, cardiovascular disease and several types of cancers [84,85,86]. A recent study concluded that the occasional substitution of meat products with ready-to-use meat alternatives (about 4–6 servings of plant-based meat per week instead of animal-derived meat) may have a beneficial impact on the gut microbiome, presumably because of the fibre content of plant-based meat alternatives [78]. Although heme iron in red meat is associated with health risks and inflammation, meat does remain a source of highly bioavailable iron as well as protein and vitamin B12 [90,91,92].

When comparing the mean nutritional composition of the present study with other studies, it is important to note the possible differences in methodology between studies (for example, the use of median values versus mean values, differences in categorisation of the products, etc.). In other studies, the mean nutritional composition of minced meat ranged between 13.7–47.3 g/100 g for protein, 5.4–10.0 g/100 g for total fat, 0.98–3.0 g/100 g for saturated fat and 0.1–2.3 g/100 g for salt [35,93,94,95]. In the present study the mean nutritional values were found within the ranges of the other studies. In other studies, the mean ranges for sausages were 13.4–18.9 g/100 g for protein, 7.9–15.0 g/100 g for total fat, 1.7–2.4 g/100 g for saturated fat and 1.5–2.3 g/100 g for salt. In the present study, the mean nutritional value of saturated fat was slightly above the range found in other studies [35,93,94,95]. In other studies, the mean nutritional composition of meatballs varies between 16.0–17.8 g/100 g protein, 5.6–9.6 g/100 g total fat, 1.1–1.2 g/100 g saturated fat and 1.4–1.8 g/100 g salt [94,95]. In the present study meatballs contained less protein, more total and saturated fat and less salt compared to the other studies. In other studies, the mean nutritional composition of burgers ranged between 9.6–16.5 g/100 g for protein, 7.2–11.3 g/100 g for total fat, 1.3–4.1 g/100 g for saturated fat and 1.4–1.6 g/100 g for salt [35,93,94,95]. In the present study, burgers contained less salt, but chicken burgers were also taken into this category, whereas the other studies (except for [94]) only included hamburgers, which might be more salty. In other studies, the mean ranges for schnitzels/nuggets were 12.3–16.7 g/100 g for protein, 10.2–12.0 g/100 g for total fat, 1.2–1.8 g/100 g for saturated fat and 1.2–1.7 g/100 g for salt [93,95]. In the present study, schnitzels/nuggets contained less total and saturated fat. In the study of [93], the mean nutritional composition of steak is 20.5 g/100 g protein, 7.5 g/100 g total fat, 1.8 g/100 g saturated fat and 1.6 g/100 g salt [93]. In the present study, steak contained less protein, more total and saturated fat and less salt. These results indicate that there are only small differences in nutritional composition between ready-to-use meat alternatives sold in different countries.

Ready-to-use meat alternatives are usually based on (a combination of) soy, wheat, pea, egg, milk protein and/or mycoprotein. In the present study mostly soy protein and wheat protein were used as main protein sources. Soy protein is considered a complete protein that meets all the essential amino acid requirements, showing a protein digestibility corrected amino acid score (PDCAAS) of 1.0 [30,76]. A characteristic of mycoprotein is that it has a high zinc but a low iron content [82]. Variation in the consumption of meat alternatives should be an essential recommendation, especially for those adopting a predominantly plant-based diet. In other studies, between 12.1% and 20.0% of the ready-to-use meat alternatives studied were fortified with iron, while between 6.0% and 24.0% were fortified in vitamin B12 [35,81,95,96]. In the current study, that number was higher, with 32.9% and 31.3% fortified with iron and vitamin B12, respectively. It should be noted that non-heme iron is added to meat alternatives, which is often absorbed less than 10% [97]. However, this indicates that many producers are aware of the standards best met by a ready-to-use meat alternative. Also, The bioavailability of vitamins and minerals in ready-to-use meat alternatives—especially those based on protein extracts—is often questioned as phytate accumulated in the protein fraction could hinder absorption [98,99]. Tempeh and mycoprotein were shown to have a low phytate content, which leads to a higher bioavailability of nutrients such as iron and zinc [82]. Future product development should focus on the optimal bioavailability of iron and vitamin B12 when developing products using protein extraction and extrusion. For individuals following a predominantly plant-based food pattern, it is even more important to look for multiple nutritional sources and to combine nutritional sources of iron with absorption-enhancing nutrients such as vitamin C while avoiding absorption inhibitors such as polyphenols [97].

According to the literature, ready-to-use meat alternatives, compared to animal-based counterparts, tend to have lower levels of total fat, saturated fat, cholesterol, and total calories but may contain less (bioavailable) protein, iron, and vitamin B12. Although some ready-to-use meat alternatives contain quite a lot of salt, they also tend to be higher in fibre and several micronutrients [41,42,43,44,45,46,47,48,49]. The results of the present study are mostly in agreement with the literature. In the present study vegetarian/vegan minced meat, hamburgers and bacon contained less total fat compared to meat (products), while vegetarian/vegan steak contained more total fat compared to meat (products). Vegetarian/vegan minced meat, vegetarian/vegan schnitzels/breaded burgers/nuggets, vegetarian/vegan hamburgers, vegetarian/vegan chicken (pieces), vegetarian/vegan gyros/shoarma/pita meat, vegetarian/vegan bacon and vegetarian/vegan sausages contained less saturated fat than meat (products). Vegetarian/vegan minced meat and vegetarian/vegan bacon contained fewer calories, while vegetarian/vegan steak contained more calories than meat (products). Vegetarian/vegan schnitzels/breaded burgers/nuggets, vegetarian/vegan cheeseburgers/schnitzels, vegetarian/vegan hamburgers, vegetarian/vegan chicken (pieces), vegetarian/vegan steak and vegetarian/vegan sausages contained less protein than meat (products), although sometimes the differences were small. The biggest difference in nutritional value in favour of the vegetarian/vegan meat alternatives was found in minced meat and bacon. The biggest difference in nutritional value against the vegetarian/vegan meat alternatives was found in steak.

Regarding planetary benefits, several studies found that the planetary impact of ready-to-use meat alternatives is lower than the planetary impact of meat, even when animal products with the lowest impact are compared with highly processed meat alternatives [1,100,101,102]. Greenhouse gas emissions could fall by 54 to 87% with a higher proportion of plant-based protein sources [8].

This study has some limitations, as it is possible that slight changes in nutritional composition occurred throughout the months of compiling the database. Presumably, there is an underestimation in the number of ready-to-use meat alternatives fortified with iron, vitamin B12 and zinc, as well as Nutriscore and Ecoscore, as not all data were declared for all ready-to-use meat alternatives. Another limitation of the study is that no information was available regarding the use of iodized salt in ready-to-use meat alternatives, although this is an important nutrient when transitioning to a more plant-based food pattern [103]. Also nutritional information about other essential nutrients like for example folate and calcium was lacking, as producers did not mention these nutrients on the packaging.

There are also a number of strengths associated with this study. A major strength of this study is that it can provide essential information for health professionals (such as dietitians) to give recommendations to consumers. The study provides an overview of mean nutritional values as well as percentages of the number of ready-to-use meat alternatives within each category that meet the norm values, which can guide consumers towards nutritionally high-quality ready-to-use meat alternatives.

In future research, more (recent) products should be added to the database, taking into account that sometimes differences in nutritional composition could be found between the values on the producer’s website, the supermarket’s website and/or the packaging. Future research should also compare the nutritional composition of unprocessed, minimally processed and ultra-processed meat alternatives to meat (products) in terms of essential amino acids, other nutrients and environmental impact.

## 5. Conclusions

Regarding the first aim of this study, it can be concluded that minced meat and pieces/strips/cubes came out as the most favourable categories regarding nutritional composition, while legume burgers/falafel were, on average, too low in protein and cheeseburgers/schnitzels, nut/seed burgers and sausages were too high in total fat according to the norm values of protein, total fat, saturated fat and salt. Regarding the second aim of this study, it can be concluded that vegetarian/vegan steak came out the least favourable compared to steak, while vegetarian/vegan minced meat and vegetarian/vegan bacon came out the most favourable in terms of nutritional composition compared to their animal-based counterparts.

The consumption of different types of plant-based protein sources should be encouraged in order to pursue a varied nutritional profile. Producers of ready-to-use meat alternatives should be advised to create products with a complete nutritional composition regarding macronutrients and micronutrients.

## Figures and Tables

**Table 1 nutrients-16-01648-t001:** General information/characteristics of the different types of ready-to-use meat alternatives.

	n	Vegan		House Brand		Gluten-Free		Frozen		Organic	
		n	%	n	%	n	%	n	%	n	%
Minced meat	27	24	88.9	8	29.6	11	40.7	3	11.1	4	14.8
Schnitzels/breaded burgers/nuggets	76	61	80.3	23	30.3	6	7.9	11	14.5	5	6.6
Vegetable burgers/balls (not breaded)	52	31	59.6	22	42.3	7	13.5	3	5.8	18	34.6
Vegetable burgers/balls (breaded)	37	14	37.8	15	40.5	0	0.0	7	18.9	10	27.0
Cheese burgers/schnitzels	33	3	9.1	12	36.4	0	0.0	0	0.0	12	36.4
(Pseudo)grain burgers	15	9	60.0	3	20.0	2	13.3	0	0.0	13	86.7
Legume burgers/falafel	60	52	86.7	25	41.7	16	26.7	2	3.3	32	53.3
Nut/seed burgers	10	6	60.0	5	50.0	3	30.0	0	0.0	6	60.0
Hamburgers/chicken burgers	45	29	64.4	15	33.3	9	20.0	6	13.3	3	6.7
Steak	12	9	75.0	5	41.7	3	25.0	1	8.3	1	8.3
Chunks/strips/cubes	77	58	75.3	28	36.4	31	40.3	10	13.0	13	16.9
Sausages	48	29	60.4	9	18.8	13	27.1	0	0.0	10	20.8
Meatballs	28	20	71.4	10	35.7	5	17.9	3	10.7	2	7.1

**Table 2 nutrients-16-01648-t002:** Differences in nutritional composition (protein, fat, saturated fat, salt, iron, vitamin B12) compared to the norm.

	Protein (g/100 g)						Total Fat (g/100 g)						Saturated Fat (g/100 g)						Salt (g/100 g)						Iron (mg/100 g)						Vitamin B12 (µg/100 g)					
	n	Min.	Max.	Mean	SD	*p*	n	Min.	Max.	Mean	SD	*p*	n	Min.	Max.	Mean	SD	*p*	n	Min.	Max.	Mean	SD	*p*	n	Min.	Max.	Mean	SD	*p*	n	Min.	Max.	Mean	SD	*p*
Minced meat	27/27	11.2	29.0	17.6	3.8	<0.001	22/27	0.5	17.0	6.7	4.8	0.001	22/27	0.1	11.3	2.1	3.0	<0.001	26/27	0.14	1.80	1.07	0.38	<0.001	10/10	2.1	10.7	4.8	3.7	0.004	10/10	0.3	0.5	0.4	0.1	0.004
Schnitzels/breaded burgers/nuggets	61/76	4.0	19.3	12.6	3.2	<0.001	50/76	2.3	18.0	9.4	3.3	0.131	75/75	0.3	2.9	1.1	0.4	<0.001	71/76	0.60	2.10	1.24	0.28	<0.001	39/39	2.1	10.7	3.3	1.9	<0.001	37/37	0.3	2.0	0.5	0.4	<0.001
Vegetable burgers/balls (not breaded)	23/52	4.6	20.0	10.2	3.9	0.698	32/52	0.2	16.1	8.4	4.0	0.007	52/52	0.0	3.7	1.2	0.8	<0.001	49/52	0.70	2.00	1.19	0.28	<0.001	11/11	2.1	3.9	2.4	0.7	0.002	11/11	0.4	0.7	0.4	0.1	0.002
Vegetable burgers/balls (breaded)	11/37	3.2	20.0	8.6	4.9	0.093	22/37	3.0	16.0	9.6	2.7	0.336	35/37	0.6	5.8	1.9	1.3	<0.001	36/37	0.00	1.70	1.04	0.33	<0.001	9/9	2.1	6.6	3.3	2.0	0.006	9/9	0.3	1.7	0.7	0.5	0.007
Cheese burgers/schnitzels	18/33	4.5	18.5	11.0	3.7	0.149	10/33	6.0	28.0	12.4	4.4	0.004	28/33	1.1	12.6	4.0	2.8	0.013	28/33	0.63	2.50	1.27	0.38	<0.001	8/8	2.1	3.2	2.3	0.4	0.008	5/5	0.4	1.0	0.6	0.3	0.039
(Pseudo)grain burgers	3/15	4.4	21.0	8.4	4.5	0.094	6/15	5.4	15.0	10.8	3.0	0.303	15/15	0.7	4.2	1.8	1.1	<0.001	14/15	0.88	4.20	1.34	0.81	0.010	1/1	3.0	3.0	3.0	/	/	0	/	/	/	/	/
Legume burgers/falafel	9/60	3.9	17.0	8.4	2.5	<0.001	34/60	2.3	17.0	10.1	3.5	0.832	60/60	0.3	4.2	1.5	0.9	<0.001	60/60	0.55	1.50	1.08	0.19	<0.001	10/10	2.1	7.0	3.5	2.0	0.004	10/10	0.4	0.9	0.5	0.2	0.004
Nut/seed burgers	6/10	7.1	18.8	12.3	4.4	0.132	0/10	12.3	23.0	18.9	4.4	<0.001	10/10	1.3	3.7	2.4	1.0	<0.001	9/10	0.84	1.63	1.10	0.27	<0.001	4/4	7.0	7.0	7.0	0.0	0.046	4/4	0.5	0.5	0.5	0.0	0.046
Hamburgers/chicken burgers	42/45	7.9	30.0	16.0	4.2	<0.001	22/45	1.5	20.0	10.4	4.9	0.577	38/44	0.5	18.0	2.5	3.1	<0.001	41/45	0.73	1.90	1.26	0.29	<0.001	22/22	2.1	10.7	4.4	2.7	<0.001	20/20	0.3	0.8	0.5	0.2	<0.001
Steak	12/12	11.2	25.5	15.3	4.0	<0.001	5/12	2.0	18.0	10.7	3.7	0.538	11/12	0.2	8.0	2.6	2.1	0.009	9/12	0.98	1.90	1.34	0.32	0.015	5/5	2.1	10.7	4.1	3.7	0.042	5/5	0.4	0.7	0.5	0.2	0.042
Chunks/strips/cubes	75/77	7.1	29.3	18.0	4.0	<0.001	63/77	0.0	19.3	7.0	4.8	<0.001	76/77	0.0	7.2	0.9	0.9	<0.001	57/77	0.79	4.05	1.50	0.55	<0.001	28/28	2.1	10.7	3.9	2.7	<0.001	29/29	0.3	2.5	0.7	0.7	<0.001
Sausages	36/48	5.4	31.3	14.8	6.6	<0.001	13/48	7.3	24.0	14.4	5.0	<0.001	43/48	0.3	13.0	2.5	2.7	<0.001	35/48	1.00	3.00	1.54	0.39	0.012	15/15	1.4	10.7	4.1	2.8	<0.001	13/13	0.4	2.0	0.7	0.6	0.001
Meatballs	25/28	8.5	22.0	15.0	3.8	<0.001	15/28	3.6	18.4	10.5	4.6	0.591	23/28	0.4	9.4	2.3	2.4	<0.001	21/26	0.65	1.80	1.37	0.28	<0.001	9/9	2.1	8.1	3.7	2.3	0.007	10/10	0.3	74.0	8.0	23.2	0.005

n: number of the total number meeting the norm value; Min.: minimum; Max.: maximum; SD: standard deviation; *p*: *p*-value. Norm values: protein (≥10 g/100 g), total fat (≤10 g/100 g), saturated fat (≤5 g/100 g), salt (≤1.625 g/100 g), iron (>0.7 mg/100 g), vitamin B12 (>0.13 µg/100 g).

**Table 3 nutrients-16-01648-t003:** Differences in Nutriscore and Ecoscore compared to the norm and differences in price per 100 g of product between the different categories.

	Nutriscore (Score at 5)						Ecoscore (Score at 5)						Price per 100 Grams of Product (Euro)					
	n	Min.	Max.	Mean	SD	*p*	n	Min.	Max.	Mean	SD	*p*	n	Min.	Max.	Mean	SD	Differences *
Minced meat	12/12	1.0	1.0	1.0	0.0	1.000	1/1	1.0	1.0	1.0	/	/	26	0.74	2.17	1.55	0.43	a
Schnitzels/breaded burgers/nuggets	30/34	1.0	3.0	1.2	0.4	0.059	2/2	1.0	1.0	1.0	0.0	1.000	66	0.75	2.38	1.60	0.45	b
Vegetable burgers/balls (not breaded)	26/29	1.0	3.0	1.1	0.4	0.102	2/2	1.0	1.0	1.0	0.0	1.000	43	0.75	2.86	1.70	0.42	c
Vegetable burgers/balls (breaded)	9/21	1.0	3.0	1.7	0.7	0.001	0	/	/	/		/	33	0.48	2.49	1.61	0.60	d
Cheese burgers/schnitzels	6/16	1.0	4.0	2.0	1.0	0.004	2/3	1.0	2.0	1.3	0.6	0.317	29	0.83	3.56	1.88	0.58	
(Pseudo)grain burgers	5/10	1.0	3.0	1.6	0.7	0.034	1/3	1.0	2.0	1.7	0.6	0.157	12	1.02	2.69	1.81	0.50	
Legume burgers/falafel	30/33	1.0	3.0	1.1	0.4	0.102	6/8	1.0	2.0	1.3	0.5	0.157	57	0.85	2.54	1.74	0.45	e
Nut/seed burgers	2/5	1.0	2.0	1.6	0.5	0.083	2/2	1.0	1.0	1.0	0.0	1.000	10	1.85	3.03	2.37	0.40	a, b, c, d, e, f, g
Hamburgers/chicken burgers	14/24	1.0	3.0	1.6	0.8	0.004	2/3	1.0	2.0	1.3	0.6	0.317	45	0.75	2.81	1.77	0.43	f
Steak	3/5	1.0	3.0	1.6	0.9	0.180	0	/	/	/	/	/	12	1.05	2.57	1.68	0.45	
Chunks/strips/cubes	23/37	1.0	4.0	1.8	1.0	<0.001	3/3	1.0	1.0	1.0	0.0	1.000	76	0.75	3.35	1.87	0.57	
Sausages	9/26	1.0	4.0	2.2	1.0	<0.001	2/6	1.0	2.0	1.7	0.5	*0.046*	46	0.92	3.00	1.86	0.47	
Meatballs	8/15	1.0	4.0	1.8	1.0	0.016	0/1	2.0	2.0	2.0	/	/	27	0.65	2.52	1.60	0.58	g

n: number; Min.: minimum; Max.: maximum; SD: standard deviation; *p*: *p*-value. * The letters represent statistically significant differences in price between categories (letters a differ significantly from each other, idem for letters b, etc.).

**Table 4 nutrients-16-01648-t004:** Most important ingredients (=top five protein sources and oils/fat sources) of the different categories of ready-to-use meat alternatives.

	Main/First Protein Source		Main/First Oil/Fat Source			Main/First Protein Source		Main/First Oil/Fat Source	
	%	Source	%	Source		%	Source	%	Source
Minced meat	51.9	Soy protein	33.3	Rapeseed oil	Nut/seed burgers	40.0	Nuts	100.0	Sunflower oil
	22.2	Pea protein	14.8	Sunflower oil		30.0	Rice		
	11.1	Mycoprotein	14.8	Coconut oil		20.0	Wheat		
	3.7	Wheat gluten/flour	37.0	No oil		10.0	Soybean		
	3.7	Pea flour							
Schnitzels/breaded burgers/nuggets	43.4	Soy protein	78.9	Sunflower oil	Hamburgers/chicken burgers	55.6	Soy protein	53.3	Sunflower oil
	13.2	Wheat protein	19.7	Rapeseed oil		11.1	Wheat protein	26.7	Rapeseed oil
	9.2	Wheat flour	1.3	Soy oil		11.1	Pea protein	13.3	Coconut oil
	7.9	Soybean				8.9	Wheat gluten	2.2	Palm fat
	5.3	Cow’s milk, goat milk				4.4	Mycoprotein, pea flour	4.4	No oil
Vegetable burgers/balls (not breaded)	32.7	Soy protein	71.2	Sunflower oil	Steak	33.3	Soy protein	91.7	Sunflower oil
	15.4	Soybean	23.1	Rapeseed oil		16.7	Mycoprotein	8.3	Coconut oil
	5.8	Oat flour	1.9	Coconut oil		16.7	Wheat protein		
	7.6	Soybean flour	3.8	No oil		16.7	Wheat gluten		
	5.8	Wheat protein				8.3	Fava bean protein; Soybean		
Vegetable burgers/balls (breaded)	32.4	Wheat flour	59.5	Sunflower oil	Chunks/strips/cubes	74.0	Soy protein	55.7	Sunflower oil
	16.2	Soy protein	32.4	Rapeseed oil		5.2	Pea protein	29.9	Rapeseed oil
	10.8	Cheese	8.1	Palm fat		5.2	Soybean	5.2	Olive oil
	10.8	Potato flour				5.2	Wheat gluten	1.3	Palm fat
	8.1	Wheat protein				3.9	Mycoprotein, wheat protein	7.8	No oil
Cheese burgers/schnitzels	42.5	Cheese (cow + goat)	75.8	Sunflower oil	Sausages	29.2	Soy protein	47.9	Rapeseed oil
	15.2	Cow’s milk	12.1	Rapeseed oil		18.8	Wheat gluten	39.6	Sunflower oil
	12.1	Wheat flour	3.0	Coconut oil; olive oil		16.7	Egg protein powder	8.3	Coconut oil
	6.1	Wheat protein, soy protein, quinoa, rice	6.1	No oil		12.5	Wheat protein	4.2	Shea butter
	3.0	Pea protein, soybean				10.4	Pea protein		
(Pseudo)grain burgers	26.7	Rice	80.0	Sunflower oil	Meatballs	50.0	Soy protein	46.5	Sunflower oil
	20.0	Wheat	6.7	Rapeseed oil; olive oil		25.0	Wheat protein	32.1	Rapeseed oil
	13.3	Quinoa, millet	6.7	No oil		21.4	Pea protein	14.3	Coconut oil
	6.7	Oat flour				3.6	Soybean	3.6	Olive oil
	6.7	Wheat gluten, wheat protein	86.7	Sunflower oil				3.6	Palm fat
Legume burgers/falafel	56.7	Chickpea	10.0	Rapeseed oil					
	11.7	Lentil	3.3	Olive oil					
	11.7	Soybean							
	5.0	Soy protein							
	3.3	Lupine bean							

% = percentage of products within each category containing the ingredient as most important/first protein source or most important/first oil/fat source.

**Table 5 nutrients-16-01648-t005:** Differences in nutritional composition (kcal, protein, total fat, saturated fat and salt) between ready-to-use meat alternatives and meat (products).

	Kcal/100 g						Protein (g/100 g)						Total Fat (g/100 g)						Saturated Fat (g/100 g)						Fibre (g/100 g)						Salt (g/100 g)					
Category	n	Min.	Max.	Mean	SD	*p*	n	Min.	Max.	Mean	SD	*p*	n	Min.	Max.	Mean	SD	*p*	n	Min.	Max.	Mean	SD	*p*	n	Min.	Max.	Mean	SD	*p*	n	Min.	Max.	Mean	SD	*p*
Minced meat (vegetarian and vegan)	27	92.0	238.0	159.0	41.4	<0.001	27	11.2	29.0	17.6	3.8	0.155	27	0.5	17.0	6.7	4.8	<0.001	27	0.1	11.3	2.1	3.0	<0.001	27	1.2	8.9	4.9	2.0	<0.001	27	0.1	1.8	1.1	0.4	0.188
Minced meat	64	103.0	273.0	211.8	44.3		64	15.4	22.0	18.2	1.7		64	1.2	22.0	14.9	5.3		64	0.6	9.3	5.7	2.2		64	0.0	1.4	0.2	0.4		64	0.1	2.4	0.9	0.6	

Schnitzels/breaded burgers/nuggets (vegetarian and vegan)	76	156.0	312.0	217.2	34.4	0.381	76	4.0	19.3	12.6	3.2	0.006	76	2.3	18.0	9.4	3.3	0.088	75	0.3	2.9	1.1	0.4	<0.001	76	1.8	7.3	4.6	1.4	<0.001	76	0.6	2.1	1.2	0.3	0.543
Schnitzels/breaded burgers/nuggets	69	110.0	308.0	211.2	46.9		69	7.0	25.0	14.4	3.5		69	1.0	20.0	10.2	4.8		68	0.3	7.3	2.5	1.5		59	0.0	4.3	0.9	0.8		65	0.1	2.0	1.2	0.3	

Cheeseburgers/schnitzels (vegetarian and vegan)	34	165.0	353.0	234.4	44.2	0.231	34	4.5	18.5	11.0	3.7	<0.001	34	6.0	28.0	12.4	4.4	0.746	34	0.8	12.6	3.9	2.8	0.081	32	0.4	8.1	3.5	1.9	<0.001	34	0.6	2.5	1.3	0.4	0.739
Cheeseburgers/schnitzels (with a meat component)	48	140.0	348.0	221.1	52.3		48	12.0	23.9	16.0	3.0		48	3.4	24.0	12.7	5.3		48	1.7	11.0	4.7	2.6		41	0.0	2.5	0.8	0.6		47	0.6	2.3	1.3	0.3	

Hamburgers (vegetarian and vegan)	29	110.0	298.0	200.4	44.8	0.119	29	11.6	30.0	17.5	4.1	0.032	29	2.1	20.0	10.7	5.0	0.004	28	0.5	18.0	3.1	3.6	<0.001	27	0.5	7.2	4.5	1.6	<0.001	29	0.7	1.9	1.3	0.3	0.224
Hamburgers	36	101.0	311.0	214.1	53.4		36	13.0	22.0	18.2	1.8		36	1.3	25.0	14.8	6.1		36	0.5	10.0	6.2	2.7		36	0.0	1.3	0.2	0.4		36	0.1	2.0	1.1	0.4	

Chicken (pieces) unbreaded and without marinade (vegetarian and vegan)	31	86.0	190.0	146.0	30.4	0.403	31	9.5	27.0	17.4	3.9	<0.001	31	1.5	10.5	5.3	2.8	0.849	31	0.3	1.2	0.7	0.3	<0.001	29	0.0	9.8	4.8	2.6	<0.001	31	0.8	2.2	1.3	0.4	<0.001
Chicken (pieces) unbreaded and without marinade	106	101.0	286.0	145.3	44.2		106	13.0	25.0	20.6	2.7		106	0.8	24.0	6.9	5.7		106	0.3	8.5	2.1	1.8		106	0.0	1.9	0.1	0.3		106	0.1	2.3	0.3	0.3	

Chicken (pieces) breaded/with marinade (vegetarian and vegan)	24	90.0	245.0	183.3	42.4	0.177	24	7.1	25.0	15.9	4.6	0.022	24	0.8	17.0	9.5	4.5	0.408	24	0.0	3.6	1.3	0.9	0.002	24	0.5	8.5	4.9	2.0	<0.001	24	0.8	2.8	1.4	0.4	0.027
Chicken (pieces) processed (breaded/with marinade)	55	99.0	284.0	168.7	44.1		55	9.0	26.0	18.1	3.6		55	1.1	19.0	8.6	4.6		55	0.3	5.6	2.3	1.4		55	0.0	3.7	0.6	0.9		55	0.3	2.4	1.2	0.4	

Steak (vegetarian and vegan)	15	76.0	258.0	184.9	50.4	<0.001	15	11.2	25.5	15.5	4.4	<0.001	15	0.3	18.0	9.1	4.7	<0.001	15	0.1	8.0	2.1	2.1	0.094	14	0.2	8.1	4.8	2.4	<0.001	15	1.0	1.9	1.3	0.3	<0.001
Steak	50	96.0	274.0	120.4	36.0		50	18.0	24.0	21.9	1.3		50	0.6	22.0	3.5	4.3		50	0.3	9.3	1.4	1.9		50	0.0	2.5	0.1	0.4		50	0.1	1.6	0.3	0.3	

Gyros/shoarma/pita meat (vegetarian and vegan)	23	108.0	306.0	160.7	54.7	0.059	23	13.0	29.3	18.6	4.6	0.418	23	0.0	16.8	6.7	4.6	0.434	23	0.0	2.0	0.8	0.4	<0.001	20	1.8	7.7	5.3	1.6	<0.001	23	1.0	1.9	1.4	0.3	0.006
Gyros/shoarma/pita meat	20	89.0	186.0	134.3	29.1		20	11.0	22.0	17.5	3.5		20	2.0	12.3	6.5	3.2		20	0.4	4.0	1.9	1.0		20	0.0	2.8	0.5	0.7		20	0.8	1.9	1.2	0.3	

Bacon (vegetarian and vegan)	11	106.0	269.0	210.8	47.7	0.002	11	14.9	21.0	17.5	1.8	0.848	11	0.6	19.3	12.7	6.5	<0.001	11	0.1	7.2	2.1	2.0	<0.001	11	1.5	6.0	3.7	1.6	<0.001	11	1.1	4.1	2.1	0.9	0.036
Bacon	85	102.0	757.0	286.8	109.4		85	2.0	55.0	18.8	6.6		85	2.3	83.0	23.2	12.1		85	1.0	30.0	8.9	4.6		85	0.0	1.0	0.0	0.2		85	0.1	8.1	2.7	1.2	

Sausages (vegetarian and vegan)	48	144.0	306.0	221.9	44.0	0.679	48	5.4	31.3	14.8	6.6	0.001	48	7.3	24.0	14.4	5.0	0.060	48	0.3	13.0	2.5	2.7	<0.001	42	0.7	7.8	3.8	1.8	<0.001	48	1.0	3.0	1.5	0.4	0.004
Sausages	91	117.0	377.0	218.4	45.9		91	4.6	20.0	16.4	2.8		91	5.0	31.0	16.1	5.2		91	1.6	12.0	6.1	2.3		91	0.0	2.0	0.2	0.4		91	0.7	2.4	1.4	0.3	

n: number of products; Min.: minimum; Max.: maximum; SD: standard deviation; *p*: *p*-value.

## Data Availability

The data presented in this study are available in Table 1, Table 2 and Table 3. The data presented in this study are available on request from the corresponding author.

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
