# Peer review of "Analysis of the Nutritional Composition of Ready-to-Use Meat Alternatives in Belgium"

_nutrients, 2024, doi:10.3390/nu16111648_

Round 1

Reviewer 1 Report

Comments and Suggestions for Authors

Dear,

Please consider the following comments when reviewing the manuscript. They can greatly enhance its structure, readability, and appropriate placement of obtained results within the topic's context.

Organise the information in a logical sequence. Start with a brief overview of 'meat alternatives' and their increasing popularity. Then, mention the database used to analyse nutritional composition and clearly explain the methods used. Consider including the products' ingredients, possibly from the manufacturer, in a supplementary table. Compare comparable meat products to which they are a substitute (in the extract, methods, results, and discussion).

Abstract:

-          Line 10: The first sentence in the text seems disconnected from the title and the broader topic of the research aim. I suggest rephrasing it to reflect the trend in the food industry towards offering more meat alternatives or promoting plant-based nutrition due to the nutritional and environmental benefits. This trend has led to new products with different dietary compositions, which raises the question of the homogeneity or heterogeneity of their nutritional values and their impact on modern nutrition.

-          Line 13: Is it relevant to check the amount of added sugar in it?

-          Line 14: Can you indicate the total number of products and specify if they are top-selling or from the biggest supermarkets so that we can evaluate the generalisation?

-          Line 15: Referring to the nutritional composition as "selected" may be more accurate since only specific nutrients were assessed. 

-          Lines 17-18: For cheeseburgers, be specific about adding parentheses based on what they are made of (cheese from coconut oil, tofu, or nuts).

-          Lines 20-21: You compared everything to "norm value"; given that it is about meat alternatives, I suggest adding a sentence comparing meat and meat products that meat alternatives should replace.

-          Line 23: .. too low  … Do you mean low or too low? Or write down how much of the "norm value" these protein products lack (8-9 g/100 g or up to 6 g/100 g)?

-          Line 23 .. and salt?

Introduction:

-          Lines 28-30: I suggest you begin by examining the cause, trends, reasons, and current status of the "meat alternative" industry instead of focusing immediately on one nutrient (protein). This approach would align better with the study's title and aim and the title. I am unsure why you are emphasizing protein if the purpose of the study is to evaluate the nutritional composition of "meat alternatives." As meat-based foods are commonly replaced with plant-based alternatives, it raises questions not only about protein, vitamin B12, and iron content as potential deficiencies and fibre and mono-saturated fats as prominent features in plant-based sources. Additionally, there may be concerns regarding excess fat and salt.

-          Lines 54-63: Consider the issue of relying on overly general classifications, which may have been accurate in the past but may be less so now, especially given that ultra-processed foods (UPFs) have a higher nutritional density than before. I recommend reviewing the following resources for more information: https://pubmed.ncbi.nlm.nih.gov/37339348/, https://pubmed.ncbi.nlm.nih.gov/36805851/

-          Lines 68-72: Could you please clarify whether the "meat alternatives" are being used to refer to people transitioning to a more plant-based diet and using these products as an easier way to do so?

-          Line 80: Lower or absent dietary cholesterol? If the topic is non-meat alternatives, it's important to note that there are different types, such as lacto, ovo, pesto, and vegan options. Additionally, within this category, there are subcategories that go beyond environmental and nutritional reasons and extend into ideology. Judge whether mentioning this in the introduction can enhance reader comprehension.

-          Line 81: You mentioned that "meat alternatives" have less bioavailable protein, iron, and vitamin B12. It would be helpful to clarify who would be affected by this, as you previously stated that these products are typically used by people who consume animal products but are transitioning to a more plant-based diet. Is it still important to consider the overall diet in this scenario?

-          Lines 88-89: Is it enough to compare the nutritional composition of "meat alternative" with "norm value" without comparing it with the meat products that replace them to get the whole/big picture?

Methods:

-          Explain precisely what "Nutriscore" and "Ecoscore" are and what they include, in short, all the necessary information.

-          Please provide the name of the used database and a reference link for the nutritional composition evaluation of the products. 

-          It is crucial to compare the nutritional composition and price of "meat alternatives" with the products they replace.

Results:

-          I suggest including the names of the products and manufacturers in the supplementary table to increase the study's credibility. Otherwise, the results could be considered fabricated.

-          Table 1: Instead of redundantly repeating the tabular results in the text, I propose you provide information about the composition of individual products below the table, for instance, whether they are made from legumes, cereals, nuts, coconut fat, etc. This approach significantly enhances the table's informativeness and clarity, instilling confidence in the reader.

-          It isn't easy to follow Table 2 and the accompanying text when one knows nothing about the composition except for a little something from the name. List the first, for example, the first five ingredients for each product.

Discussion:

-          The discussion needs to include the reasons for the obtained results. For instance, it needs to explain why the bean burger has insufficient protein or why certain products have more total fats and their source. Additionally, the text raises the question of salt: Is it iodized? If so, should it be used?

-          Lines 223-230: Please consider adding the recommended sources suggested in the Introduction if they would improve or expand the discussion about the semi-obsolete method of categorizing foods solely based on the extent of processing rather than considering their nutritional composition and related scientific evidence.

-          Line 257: Also, please include a discussion on the nutritional composition and health benefits (with references) of "meat alternatives" such as soy tofu, tempeh (made from soy and other legumes), seitan, and similar products. This will provide a balanced comparison between meat products and different meat alternatives, which will help put them into a fair context.

-          Lines 293-296: Please clarify and elaborate on your point about "meat alternatives" contain less bioavailable protein, iron, and vitamin B12? It would be helpful to understand why this is important, whether it is crucial for whom, and whether it is just a minor concern in the context of a varied omnivorous diet. In addition .. Could you please help me understand if the problem with phytates results in impaired mineral absorption when using a "meat alternative"? I would like to know if using these products completely changes the diet of omnivorous individuals. Also, are legumes recommended in an omnivorous diet that people often need to consume more of? This is also a challenge in Belgium, so I expect a more balanced scientific interpretation and discussion (https://pubmed.ncbi.nlm.nih.gov/21443812/; https://www.ncbi.nlm.nih.gov/pmc/articles/PMC3436705/; https://www.brusselstimes.com/55552/flemish-millennials-told-to-eat-more-vegetables  … Furthermore, iron can be found in two forms: heme and non-heme. Heme iron is only found in animal products, while non-heme iron is found in plant and animal products. Expand on this fact in the discussion.

-          I suggest having a separate section titled 'Strengths and Limitations'.

-          One limitation of the analysis is that it didn't include essential nutrients like fibre, vitamin C, folate, calcium, and zinc.

-          Future research should also compare the nutritional composition of unprocessed, partially processed, and ultra-processed meat alternatives to meat in terms of essential amino acids, other nutrients, and environmental impact. Iodized salt may need to be used for this purpose .. possibly changing the classification of UPFs?

Conclusion:

-          Please provide an explanation for the results that were obtained. 

Author Response

Rebuttal manuscript: “Analysis of the nutritional composition of ready-to-use meat alternatives in Belgium”

Section: Nutrition and Public Health

Special Issue: Nutritional Value of Meat Alternatives and Their (Possible) Contribution to Human Health

First of all, the authors would like to thank the reviewers for taking the time to thoroughly go through the manuscript and provide valuable feedback, all in order to make the paper stronger. Several changes in all sections were made and an attempt is made to describe these changes as clearly as possible in this rebuttal. Since a lot of changes were made to the paper, all changes were clearly highlighted in yellow in the paper itself.

Reviewer 1

Dear,

Please consider the following comments when reviewing the manuscript. They can greatly enhance its structure, readability, and appropriate placement of obtained results within the topic's context.

Organise the information in a logical sequence. Start with a brief overview of 'meat alternatives' and their increasing popularity. Then, mention the database used to analyse nutritional composition and clearly explain the methods used. Consider including the products' ingredients, possibly from the manufacturer, in a supplementary table. Compare comparable meat products to which they are a substitute (in the extract, methods, results, and discussion).

The authors would like to thank the reviewer, as they agree that re-writing the information in a more logical sequence improved readability. Therefore, (1) the introduction was structured slightly differently, (2) the method was more extensively described, two tables of results (table 4 with ingredients (main protein sources and main oils/fat sources) and table 5 with the comparison between ready-to-use meat alternatives and meat (products)) were added, and (3) the discussion was expanded taking into account the reviewers' feedback and the additional results of the analyses that were performed.

Abstract:

Line 10: The first sentence in the text seems disconnected from the title and the broader topic of the research aim. I suggest rephrasing it to reflect the trend in the food industry towards offering more meat alternatives or promoting plant-based nutrition due to the nutritional and environmental benefits. This trend has led to new products with different dietary compositions, which raises the question of the homogeneity or heterogeneity of their nutritional values and their impact on modern nutrition.

Line 13: Is it relevant to check the amount of added sugar in it?

Line 14: Can you indicate the total number of products and specify if they are top-selling or from the biggest supermarkets so that we can evaluate the generalisation?

Line 15: Referring to the nutritional composition as "selected" may be more accurate since only specific nutrients were assessed.

Lines 17-18: For cheeseburgers, be specific about adding parentheses based on what they are made of (cheese from coconut oil, tofu, or nuts).

Lines 20-21: You compared everything to "norm value"; given that it is about meat alternatives, I suggest adding a sentence comparing meat and meat products that meat alternatives should replace.

Line 23: .. too low  … Do you mean low or too low? Or write down how much of the "norm value" these protein products lack (8-9 g/100 g or up to 6 g/100 g)?

Line 23 .. and salt?

Since the paper underwent major changes, the authors decided to completely rewrite the abstract taking into account the reviewers' feedback and the additional statistical analyses (= two extra tables: table 4 and table 5) that were performed. (page 1 in the paper)

Introduction:

Lines 28-30: I suggest you begin by examining the cause, trends, reasons, and current status of the "meat alternative" industry instead of focusing immediately on one nutrient (protein). This approach would align better with the study's title and aim and the title. I am unsure why you are emphasizing protein if the purpose of the study is to evaluate the nutritional composition of "meat alternatives." As meat-based foods are commonly replaced with plant-based alternatives, it raises questions not only about protein, vitamin B12, and iron content as potential deficiencies and fibre and mono-saturated fats as prominent features in plant-based sources. Additionally, there may be concerns regarding excess fat and salt.

We would like to thank you for this comment which helped us to improve the introduction. The first paragraph was updated taking into account the feedback of the reviewer. (page 1 in the paper)

Lines 54-63: Consider the issue of relying on overly general classifications, which may have been accurate in the past but may be less so now, especially given that ultra-processed foods (UPFs) have a higher nutritional density than before. I recommend reviewing the following resources for more information: https://pubmed.ncbi.nlm.nih.gov/37339348/, https://pubmed.ncbi.nlm.nih.gov/36805851/

The authors want to thank the reviewer for proposing these two extra references, as they are very valuable in making the introduction of the paper stronger and in creating more fine distinction regarding ultra-processed foods. The authors strongly agree that nuance is incredibly important to distinguish between ultra-processed foods that carry negative health effects and (ultra-)processed foods with a higher nutrient density. Therefore the authors added an extra sentence with three extra references in the introduction. (page 2 in the paper)

Lines 68-72: Could you please clarify whether the "meat alternatives" are being used to refer to people transitioning to a more plant-based diet and using these products as an easier way to do so?

Several studies have shown that consumers who favor the consumption of meat prefer meat alternatives that are similar to meat; contrastingly, the more people are in favor of the consumption of meat alternatives, the less they want these alternatives to resemble meat. This means that individuals opting for a vegetarian or vegan food pattern do not need meat alternatives that resemble the organoleptic and sensory properties of meat as much, whereas omnivores and flexitarians prefer meat alternatives that approximate meat in all respects. This information and additional references are presented in the introduction. (pages 1 and 2 in the paper)

Line 80: Lower or absent dietary cholesterol? If the topic is non-meat alternatives, it's important to note that there are different types, such as lacto, ovo, pesto, and vegan options. Additionally, within this category, there are subcategories that go beyond environmental and nutritional reasons and extend into ideology. Judge whether mentioning this in the introduction can enhance reader comprehension.

The authors agree that the cholesterol level of the ready-to-use meat alternatives is dependent on the fact whether there are still animal-based ingredients present in the product. Totally plant-based meat alternatives don’t contain cholesterol, as vegetarian products might have a (very) small quantity of cholesterol. The authors agree that there could be a variety of reasons why people tend to choose for meat alternatives. This information about different types of food patterns and reasons for choosing a specific type of food pattern is added in the first paragraph of the introduction. (page 1 in the paper)

Line 81: You mentioned that "meat alternatives" have less bioavailable protein, iron, and vitamin B12. It would be helpful to clarify who would be affected by this, as you previously stated that these products are typically used by people who consume animal products but are transitioning to a more plant-based diet. Is it still important to consider the overall diet in this scenario?

The authors agree that the points of interest of meat alternatives (such as less bioavailable protein, less iron, less vitamin B12) are of less importance for those adopting a varied omnivore food pattern. It is more important for those adopting a totally plant-based diet. Therefore a sentence was added in the introduction to emphasize that variety in (plant-based) protein sources remains important and that the total food pattern should be taken into account. (page 2 in the paper)

Lines 88-89: Is it enough to compare the nutritional composition of "meat alternative" with "norm value" without comparing it with the meat products that replace them to get the whole/big picture?

The authors agree that comparing the nutritional composition of ready-to-use meat alternatives with meat (products) would provide valuable insights. Therefore the authors performed additional statistical analyses in SPSS to compare the nutritional composition (kcal, protein, total fat, saturated fat, fibre and salt) between ready-to-use meat alternatives and meat (products) (specifically their animal-based counterparts). These results are shown in table 5. The authors also supplemented the method section with information regarding the extra statistical analyses which were performed. (pages 3, 4, 5, 6, 9, 10, 11 and 13 in the paper)

Methods:

Explain precisely what "Nutriscore" and "Ecoscore" are and what they include, in short, all the necessary information.

The authors agree that the information/definition about “Nutriscore” and “Ecoscore” was not mentioned in the method section. Therefore, the authors added this information in the method section. (page 3 in the paper)

Please provide the name of the used database and a reference link for the nutritional composition evaluation of the products.

The authors added a supplementary file in Excel with (1) the nutritional composition, Nutriscore, Ecoscore, price and brand of the ready-to-use meat alternatives and (2) the nutritional composition of meat (products)/their animal-based counterparts.

It is crucial to compare the nutritional composition and price of "meat alternatives" with the products they replace.

The authors agree that comparing the nutritional composition of ready-to-use meat alternatives with meat (products) would provide valuable insights. Therefore the authors performed additional statistical analyses in SPSS to compare the nutritional composition (kcal, protein, total fat, saturated fat, fibre and salt) between ready-to-use meat alternatives and meat (products) (specifically their animal-based counterparts). These results are shown in table 5. The authors also supplemented the method section with information regarding the extra statistical analyses which were performed. (pages 3, 4, 5, 6, 9, 10, 11 and 13 in the paper)

Results:

I suggest including the names of the products and manufacturers in the supplementary table to increase the study's credibility. Otherwise, the results could be considered fabricated.

The authors added a supplementary file in Excel with (1) the nutritional composition, Nutriscore, Ecoscore, price and brand of the ready-to-use meat alternatives and (2) the nutritional composition of meat (products)/their animal-based counterparts.

Table 1: Instead of redundantly repeating the tabular results in the text, I propose you provide information about the composition of individual products below the table, for instance, whether they are made from legumes, cereals, nuts, coconut fat, etc. This approach significantly enhances the table's informativeness and clarity, instilling confidence in the reader.

The authors agree that providing information about the ingredients would give an explanation to the results shown in table 2. Therefore the authors added an extra table (= table 4) with information about the ingredients of the products. A top five of first protein sources and first oils/fat sources was added with additional information about the percentage of products per category containing that ingredient as respectively first protein source and first oil/fat source. (pages 5, 8 and 10 in the paper)

It isn't easy to follow Table 2 and the accompanying text when one knows nothing about the composition except for a little something from the name. List the first, for example, the first five ingredients for each product.

The authors agree that providing information about the ingredients would give an explanation to the results shown in table 2. Therefore the authors added an extra table (= table 4) with information about the ingredients of the products. A top five of first protein sources and first oils/fat sources was added with additional information about the percentage of products per category containing that ingredient as respectively first protein source and first oil/fat source. (pages 5, 8 and 10 in the paper)

Discussion:

The discussion needs to include the reasons for the obtained results. For instance, it needs to explain why the bean burger has insufficient protein or why certain products have more total fats and their source. Additionally, the text raises the question of salt: Is it iodized? If so, should it be used?

The authors agree that in the first version of the paper, too little explanations for the results were present in the discussion. Therefore, the authors added additional information regarding the explanations of the results in Table 2 (using the ingredients listed in Table 4), as well as an explanation for the newly added Table 5 (= comparison ready-to-use meat alternatives with meat (products)/their animal-based counterparts).

Unfortunately, there was no information available on the package regarding the use of iodized salt. However, the authors agree that this is an important consideration in the transition towards a more plant-based diet. Therefore, this information was added as a limitation of the present study. (pages 8, 9, 10-13 in the paper)

Lines 223-230: Please consider adding the recommended sources suggested in the Introduction if they would improve or expand the discussion about the semi-obsolete method of categorizing foods solely based on the extent of processing rather than considering their nutritional composition and related scientific evidence.

The authors agree that the extra references mentioned in the introduction can make the discussion stronger, as nuance is needed when discussing about different types of ultra-processed foods. Therefore the authors added the three extra references which were mentioned in the introduction also in the discussion. (pages 1 and 11 in the paper)

Line 257: Also, please include a discussion on the nutritional composition and health benefits (with references) of "meat alternatives" such as soy tofu, tempeh (made from soy and other legumes), seitan, and similar products. This will provide a balanced comparison between meat products and different meat alternatives, which will help put them into a fair context.

The authors agree that information about the health effects of legumes (and tofu, tempeh), but also seitan should be added to the discussion in order to provide a balanced comparison between different types of meat alternatives. Therefore the authors added information with accompanying references about the positive health effects of unprocessed and minimally processed plant-based protein sources such as legumes (and tofu, tempeh) and seitan. (pages 10 and 11 in the paper)

Lines 293-296: Please clarify and elaborate on your point about "meat alternatives" contain less bioavailable protein, iron, and vitamin B12? It would be helpful to understand why this is important, whether it is crucial for whom, and whether it is just a minor concern in the context of a varied omnivorous diet. In addition .. Could you please help me understand if the problem with phytates results in impaired mineral absorption when using a "meat alternative"? I would like to know if using these products completely changes the diet of omnivorous individuals. Also, are legumes recommended in an omnivorous diet that people often need to consume more of? This is also a challenge in Belgium, so I expect a more balanced scientific interpretation and discussion (https://pubmed.ncbi.nlm.nih.gov/21443812/; https://www.ncbi.nlm.nih.gov/pmc/articles/PMC3436705/; https://www.brusselstimes.com/55552/flemish-millennials-told-to-eat-more-vegetables  … Furthermore, iron can be found in two forms: heme and non-heme. Heme iron is only found in animal products, while non-heme iron is found in plant and animal products. Expand on this fact in the discussion.

The authors want to thank the reviewer for this valuable suggestion. In the discussion, the authors added information regarding (1) which group of people should pay extra attention to the intake of certain micronutrients such as iron and zinc, (2) information about the health benefits of legumes, (3) a discussion about bioavailability of heme iron versus non-heme iron. Also one of the proposed references was added in the discussion. (pages 1 and 10-12 in the paper)

I suggest having a separate section titled 'Strengths and Limitations'.

The authors don’t really know whether this is allowed in the journal Nutrients. The authors will leave it this way for now and check with the editor whether it is allowed to have a separate section titled ‘strengths and limitations’.

One limitation of the analysis is that it didn't include essential nutrients like fibre, vitamin C, folate, calcium, and zinc.

Unfortunately, in Belgium a lot of essential nutrients are not mentioned on the packaging, such as for example vitamin C. Fibre is a nutrient which is obligatory mentioned on the packaging and therefore the authors added fibre in the statistical analyses when ready-to-use meat alternatives were compared with meat (products)/their animal-based counterparts. Zinc was also a nutrient which was collected in the database, but unfortunately this information was available on the packaging of only a few products. The authors want to thank the reviewer for this comment, as they added this as a limitation of the study. (page 13 in the paper)

Future research should also compare the nutritional composition of unprocessed, partially processed, and ultra-processed meat alternatives to meat in terms of essential amino acids, other nutrients, and environmental impact. Iodized salt may need to be used for this purpose .. possibly changing the classification of UPFs?

The authors agree that this is relatively uncharted territory in research and added this suggestion to the paragraph regarding future research. (page 13 in the paper)

Conclusion:

Please provide an explanation for the results that were obtained.

The authors added an explanation for the results that were obtained in the discussion section and expanded the conclusion with the results of the extra performed statistical analyses. (pages 13 and 14 in the paper)

Extra comment: The authors are aware that there might be some grammar and language errors, as the English language is not the native language. Once the paper's content is approved, the authors are going to have the paper proofread for language errors.

Reviewer 2 Report

Comments and Suggestions for Authors

1.       What is the difference of vegetarian and vegan. I think the key words should be reconsidered by adding some words about nutrition.

2.       How to select the nutritional composition ? how about the carbohydrate? dietary fiber?, etc.

3.       Some meat alternative products are not ready-to-use. Please consider some more.

4.       Line 286, milk protein???

5.       Please take some attention to the grammar errors.

Comments on the Quality of English Language

1.       What is the difference of vegetarian and vegan. I think the key words should be reconsidered by adding some words about nutrition.

2.       How to select the nutritional composition ? how about the carbohydrate? dietary fiber?, etc.

3.       Some meat alternative products are not ready-to-use. Please consider some more.

4.       Line 286, milk protein???

5.       Please take some attention to the grammar errors.

Author Response

Rebuttal manuscript: “Analysis of the nutritional composition of ready-to-use meat alternatives in Belgium”

Section: Nutrition and Public Health

Special Issue: Nutritional Value of Meat Alternatives and Their (Possible) Contribution to Human Health

First of all, the authors would like to thank the reviewers for taking the time to thoroughly go through the manuscript and provide valuable feedback, all in order to make the paper stronger. Several changes in all sections were made and an attempt is made to describe these changes as clearly as possible in this rebuttal. Since a lot of changes were made to the paper, all changes were clearly highlighted in yellow in the paper itself.

Reviewer 2

What is the difference of vegetarian and vegan. I think the key words should be reconsidered by adding some words about nutrition.

The authors agree that this kind of important information is currently lacking in the introduction. Therefore the authors added some information about types of food patterns in the introduction. Also in the method section the terms vegetarian and vegan are explained in order to indicate that both vegetarian and vegan/totally plant-based ready-to-use meat alternatives are included in the present study. (pages 1 and 3 in the paper)

How to select the nutritional composition ? how about the carbohydrate? dietary fiber?, etc.

The authors also gathered information about carbohydrates and fibre, but selected the nutritional composition values of ready-to-use meat alternatives taking into account the norm values (protein (≥10 g/100 g), total fat (≤10 g/100 g), saturated fat (≤5 g/100 g), salt (≤1.625 g/100 g), iron (>0.7 mg/100 g) and vitamin B12 (>0.13 µg/100 g)) developed by the Belgian professional association of dietitians. The authors agree that this is a fair point by the reviewer, and therefore additional statistical analyses were performed in SPSS to compare the nutritional composition of ready-to-use meat alternatives with meat (products)/their animal-based counterparts (= table 5), also taking into account fibre content, as this is a very important nutrient for health. (pages 3, 5, 6, 9)

Some meat alternative products are not ready-to-use. Please consider some more.

The authors acknowledge that nowadays there is a great variety of meat alternative products (such as legumes (and tofu, tempeh), seitan and more ready-to-use meat alternatives. However, the scope of the present study was to evaluate the nutritional composition of ready-to-use meat alternatives (= last generations meat alternatives), as the unprocessed and minimally processed have already received more attention in literature while there are still a lot of questions about the health effects of ready-to-use meat alternatives. Although, the authors want to thank the reviewer for this valuable comment, as they provided an overview of health benefits of unprocessed and minimally processed meat alternatives in the discussion in order to provide a balanced comparison between different types of meat alternatives. (pages 10 and 11 in the paper)

Line 286, milk protein???

Because of the fact that also vegetarian ready-to-use meat alternatives were included in the present study, milk protein or cheese was also a possible protein source of the ready-to-use meat alternatives. This information became more clear as the authors added an extra table (= table 4) with results regarding the most important/first protein source and the most important/first oil/fat source per category of ready-to-use meat alternative.

Please take some attention to the grammar errors.

The authors are aware that there might be some grammar and language errors, as the English language is not the native language. Once the paper's content is approved, the authors are going to have the paper proofread for language errors.

Round 2

Reviewer 1 Report

Comments and Suggestions for Authors

Dear,

I congratulate the authors for the significant corrections and additions. 

The manuscript is now incomparably better and, with minor corrections, suitable for publication. 

Before submitting it, I advise the authors to carefully review the text for any inaccuracies, vagueness, disproportionate generalizations, or claims/statements that are not true or lack references.

With respect,

Abstract:

-            Line 11: List of reasons for using alternative meat substitutes. The phrase "different reasons" is not a sufficient explanation (from environmental, health, economic and ethical reasons ..)

-            Line 12: .. with meat or meat products or meat and meat products?

Introduction:

-            Lines 93-95: Add a reference for this claim; if it is common, delimit it since the paragraph combines unrelated statements.

These points of interest should be considered by consumers, especially high risk groups such as children, pregnant women and elderly, as variation in (plant-based) protein sources deserves recommendation and a qualitative food pattern should be pursued.

Results:

-            In tables, you should use a comma (,) to separate whole numbers, but I think it is appropriate to use a period (.). Tables 2 and 3 are partially opaque. I would appreciate it if you made them more readable.

-            Lines 332-333: Health claims not proven and verified are not allowed, except maybe for soy, which is associated with LDL cholesterol and, therefore, heart health. Please rephrase the claim for seitan to explain how its composition and nutritional content relate to specific health benefits.

Seitan helps in bowel movement, increases gut microbiota, decreases serum cholesterol levels, reduces postprandial blood glucose level and lowers the risk of cancer and colitis [77]. .. increases gut microbiota – diversity??

-            Lines 336-337: What are the "health disadvantages"?

.. eight out of the 10 categories of meat (products) in the present 366 study are ultra-processed, which are associated with health disadvantages [84-86].

-            Lines 403-404: What does a favourable amino acid profile mean? Does it indicate that it contains all the essential amino acids? All plant-based foods do! Or does it mean that it contains an adequate amount of them in a typical serving? What defines adequacy? Use precise terminology.  … You wrote like that higher zinc and iron content in this product may have adverse effects, similar to the issues with iodine in some sea vegetables or selenium in Brazil nuts. Please explain why or correct the statement if it is inaccurate.

Soy protein possesses a favourable amino acid profile, while mycoprotein has a high zinc but low iron content [82]

-            Lines 415-416: Do you mean "nutrient" specifically (per se - all of them), or are you referring to certain minerals? Please be specific.

Tempeh and mycoprotein showed to have a low phytate content, which leads to a higher bioavailability of nutrients [82].

Comments on the Quality of English Language

The English used could be more precise, with fewer repetitive expressions and more scientific language usage.

Author Response

Rebuttal manuscript: “Analysis of the nutritional composition of ready-to-use meat alternatives in Belgium”

Section: Nutrition and Public Health

Special Issue: Nutritional Value of Meat Alternatives and Their (Possible) Contribution to Human Health

The authors would like to thank the reviewer again for taking the time to thoroughly go through the manuscript and provide valuable feedback, all in order to make the paper even stronger. Several changes in all sections were made and an attempt is made to describe these changes as clearly as possible in this rebuttal. All changes were clearly highlighted in yellow in the paper itself.

Reviewer 1

Dear,

I congratulate the authors for the significant corrections and additions.

The manuscript is now incomparably better and, with minor corrections, suitable for publication.

Before submitting it, I advise the authors to carefully review the text for any inaccuracies, vagueness, disproportionate generalizations, or claims/statements that are not true or lack references.

With respect,

The authors would like to thank the reviewer for once again going through the paper thoroughly and for suggesting modifications.

Abstract:

Line 11: List of reasons for using alternative meat substitutes. The phrase "different reasons" is not a sufficient explanation (from environmental, health, economic and ethical reasons ..)

The authors agree with that more information would be useful in this sentence. However, the abstract should only contain 200 words and currently contains 200, so unfortunately there is no place left to include the reasons. If it had been possible in terms of word count, the authors would surely have included the reasons.

Line 12: .. with meat or meat products or meat and meat products?

The authors added “products” in brackets, as in the remainder of the paper (introduction, method, results, discussion and conclusion).

Introduction:

Lines 93-95: Add a reference for this claim; if it is common, delimit it since the paragraph combines unrelated statements.

These points of interest should be considered by consumers, especially high risk groups such as children, pregnant women and elderly, as variation in (plant-based) protein sources deserves recommendation and a qualitative food pattern should be pursued.

The authors modified the sentence and added two references. (page 2 in the paper)

Although plant-predominantly and totally plant-based food patterns are appropriate for all stages of the life cycle (including pregnancy and lactation, childhood, at older ages, and for athletes), these points of interest should be taken into account [1, 6]. A well-balanced plant-based food pattern and the regular use of fortified foods and/or supplements should be pursued [1, 6].

Results:

In tables, you should use a comma (,) to separate whole numbers, but I think it is appropriate to use a period (.). Tables 2 and 3 are partially opaque. I would appreciate it if you made them more readable.

The authors have replaced the commas in the tables with full stops/dots. The authors do not know why the tables were not clear/visible, as they were simply copied and pasted from the Excel file. To ensure good visibility of the tables, the tables in Excel were also uploaded in the ZIP file of this rebuttal/revision.

Lines 332-333: Health claims not proven and verified are not allowed, except maybe for soy, which is associated with LDL cholesterol and, therefore, heart health. Please rephrase the claim for seitan to explain how its composition and nutritional content relate to specific health benefits.

Seitan helps in bowel movement, increases gut microbiota, decreases serum cholesterol levels, reduces postprandial blood glucose level and lowers the risk of cancer and colitis [77]. .. increases gut microbiota – diversity??

The authors rephrased the claim for seitan to explain how its composition and nutritional content relate to specific health benefits. (page 12 in the paper)

The composition and nutritional profile of seitan, which is low in (saturated) fat and calories, contains complex carbohydrates and is high in plant protein, helps in bowel movement and can lead to an increase in gut microbiota diversity, a decrease in serum cholesterol levels, a reduction in postprandial blood glucose level and a decreased risk of cancer and colitis [77].

Lines 336-337: What are the "health disadvantages"?

.. eight out of the 10 categories of meat (products) in the present 366 study are ultra-processed, which are associated with health disadvantages [84-86].

The authors added the health disadvantages, such as an increased risk of type 2 diabetes, cardiovascular disease and several types of cancers with accompying references. (page 13 in the paper)

Lines 403-404: What does a favourable amino acid profile mean? Does it indicate that it contains all the essential amino acids? All plant-based foods do! Or does it mean that it contains an adequate amount of them in a typical serving? What defines adequacy? Use precise terminology.  … You wrote like that higher zinc and iron content in this product may have adverse effects, similar to the issues with iodine in some sea vegetables or selenium in Brazil nuts. Please explain why or correct the statement if it is inaccurate.

Soy protein possesses a favourable amino acid profile, while mycoprotein has a high zinc but low iron content [82]

The authors agree with the finding that all plant-based foods contain all essential amino acids and have therefore made the sentence regarding soy more specific. (page 13 in the paper)

Soy protein is considered a complete protein that meets all the essential amino acid requirements, showing a protein digestibility corrected amino acid score (PDCAAS) of 1.0 [30, 76]. Characteristic of mycoprotein is that it has a high zinc but a low iron content [82].

Lines 415-416: Do you mean "nutrient" specifically (per se - all of them), or are you referring to certain minerals? Please be specific.

Tempeh and mycoprotein showed to have a low phytate content, which leads to a higher bioavailability of nutrients [82].

The authors added the nutrients that were the subject of the relevant study/reference, namely iron and zinc. (page 14 in the paper)